# Ideas and perspectives: Synergies from co-deployment of negative emission technologies

Thorben Amann[1] and Jens Hartmann[1]

[1]Institute for Geology, Center for Earth System Research and Sustainability, Universität Hamburg, Germany

*Correspondence to*: Thorben Amann (science@thorbenamann.de)

**Abstract.** Numerous publications propose the deployment of negative emission technologies, which intend to actively remove $CO_2$ from the atmosphere with the goal to reach the 1.5° target as discussed by the IPCC. The increasing amount of scientific studies on the individual potential of different envisaged technologies and methods indicates, that no single method has enough capacities to mitigate the issue by
itself. It is thus expected that technology portfolios are deployed. As some of them utilize the same environmental compartment, co-deployment effects are expected. Those effects are particularly important to evaluate with respect to additional $CO_2$ uptake. Considering soils as one of the main affected compartments, we see a plethora of processes which can positively benefit from each other, cancelling out negative side effects or increasing overall $CO_2$ sequestration potentials. To derive more reliable estimates of
negative emission potentials and to evaluate common effects on global carbon pools, it is now necessary to intensively study interrelated effects of negative emission technology deployment while minimizing side effects.

### Introduction

As global mean temperatures are projected to increase further, strategies to mitigate climate change in
time by decreasing $CO_2$ emissions seem to slowly take effect (Jackson et al., 2015). Some $CO_2$ emission pathways include negative carbon emission strategies (Fuss et al., 2014; Fuss et al., 2016; Rogelj et al., 2018), that essentially capture $CO_2$ from the atmosphere in different ways, storing them, in the long term, as $CO_2$ molecules, or as organic and inorganic compounds (Caldeira et al., 2013). All discussed options and technologies have yet to reach the large-scale deployment stage (Minx et al., 2018; Nemet et al.,
2018). Most technologies are immature, lacking deep research on the global potential, technical feasibility, economics of deployment, and especially an assessment of the expected side effects (National Research Council, 2015; Fuss et al., 2018).
The proposed negative emission technologies (NET) encompass highly technical engineering solutions as well as methods that rely on natural processes, like growth of biomass (*e.g.,* bioenergy with carbon
capture and storage (BECCS), and afforestation), soil carbon increase, biochar, and chemical weathering (*e.g.,* Enhanced Weathering (EW) and ocean liming). As these methods are aimed to be integrated in the global biogeochemical cycle and will redistribute carbon between reservoirs (Keller et al., 2018), their interaction is inevitable if NETs are deployed at the largest scale. As such, it must be assessed how the co-deployment of NETs will affect the individual and overall efficiency, since until now, publications

focus generally on single NETs, disregarding any effects on concurrent deployment of additional technologies.

Findings from NET specific literature suggest that assessing the effects of combined NET rollout is advisable and future research should include $CO_2$ sequestration enhancing side effects that could increase the overall potential of NETs. However, the principal interaction between proposed methods needs to be studied in detail beforehand, to understand effects on the carbon pools (Fig. 1).

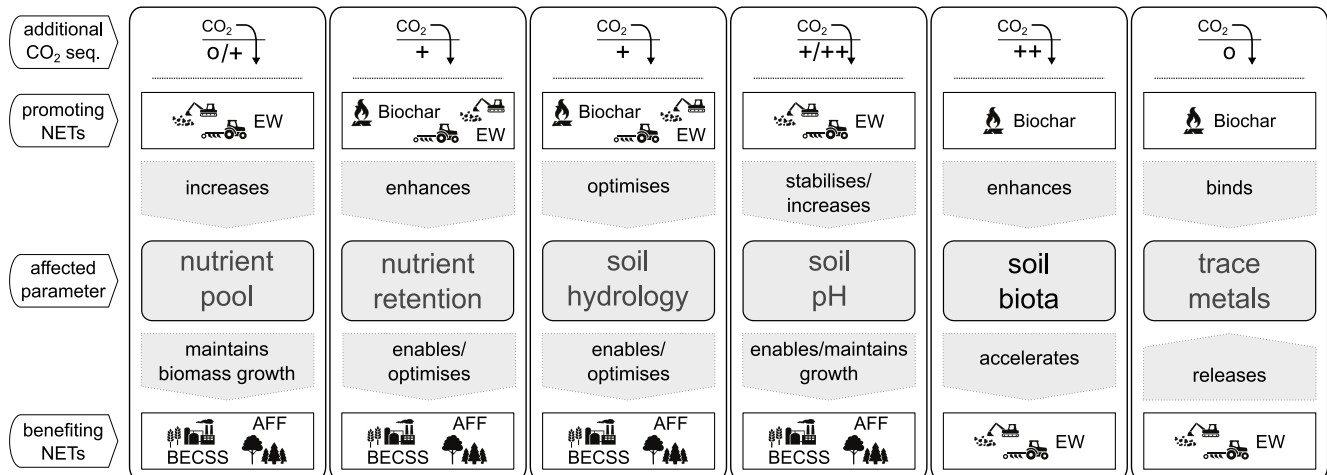

**Fig. 1: Overview of effects from combining land-based negative emission technologies (bioenergy production coupled with carbon capture and storage (BECCS), afforestation (AFF), Enhanced Weathering (EW), and biochar). The additional $CO_2$ sequestration is a qualitive estimate based on the author's personal assessment. Technology symbols courtesy of W. Lamb (MCC Berlin).**

While biomass-based NETs like afforestation and BECCS are widely discussed, EW is under-represented in this discussion (Minx et al., 2018). EW relies on the uptake of $CO_2$ via dissolution of minerals based on the natural process of chemical weathering. EW is facilitated by the application of finely ground rock on (agricultural) land, preferably in areas with elevated temperatures and rainfall. The resources for this NET have to be mined and, depending on the type and scale of rollout, the extraction of material can lead the creation of extensive mining areas. However, under a sustainable approach, affected environments could later be used to create biodiversity hotspots (e.g. Tropek et al., 2010; Benes et al., 2003).

It is unavoidable that the intended $CO_2$ sequestration effect by weathering is naturally accompanied by the release of elements with consequences for the environment (Kantola et al., 2017) and consequently the involved carbon pools. The release of elements that are important plant nutrients (*e.g.*, potassium, phosphorus, magnesium) can be beneficial for additional $CO_2$ sequestration via organic carbon formation. In addition, the soil hydrology can be improved, and cation exchange capacity increased under optimal grain size distribution and mineral selection. In contrast, effects of potentially harmful trace element release (by choosing less suitable material) might be needed to be alleviated. However, an integrated framework to achieve optimization of interrelated effects between land-based NETs has yet to be developed, specifically for a global scale management of carbon pools.

To tackle the issue of climate change with negative carbon emission strategies on a global and comprehensive scale, it seems advisable to consider all proposed terrestrial biomass-based NETs, like *e.g.*, BECSS, afforestation, and biochar to explore synergistic effects (Fig. 1). A scenario can be envisioned,

where rock powder and biochar are applied to agricultural land, which is used for bioenergy plant production (for further use in BECSS technology). Rock material would release geogenic nutrients and biochar could enhance the release of nutrients (Atkinson et al., 2010) and the overall crop productivity (Jeffery et al., 2011).

In combination with envisioned and deployed afforestation efforts, which often take place in tropical areas with depleted soils (Nilsson and Schopfhauser, 1995; Grainger, 1988; Zomer et al., 2008), rock powder deployment for EW could be an added, if not essential, benefit. The low capacity of these soils to retain highly soluble industrial fertilisers suggests the use of other forms of slow release fertiliser, like rock dust as a complement (Leonardos et al., 1987; Manning, 2015), or new emerging rock-based ferti-

lizers (Ciceri and Allanore, 2019), which can, as a side effect, increase the retention of industrial fertilizers, that may still be needed in addition. The ultimate need for an intense management and design of a suitable soil to supply suitable conditions for tree growth can be deduced from a published extreme scenario, which envisions large scale afforestation of deserts (Ornstein et al., 2009).

It seems advisable to combine proposed NET methods to achieve an optimal carbon pool management

for negative emissions and ensure food security over centuries at the global scale. To achieve this, interdisciplinary efforts are necessary (Fig. 1) and some of the key issues are reviewed here to point out the future research directions.

**Nutrient pool**

Increasing atmospheric $CO_2$ concentrations and an increasing world population will lead to challenges in

the nutritional supply for large parts of Earth's population (Smith and Myers, 2018; Myers et al., 2014). In combination with partly declining resources of natural mineral fertilisers (Manning, 2015 and Suppl. Mat. S1), alternative nutrient supplies, *i.e.*, from rock products, are of high interest (Ciceri and Allanore, 2019). This idea has been discussed earlier (Van Straaten, 2006; van Straaten, 2002) and was recently revived in the context of EW (Beerling et al., 2018; Hartmann et al., 2013). However, this issue extends

further, if biomass-based NETs are considered for large scale deployment. While nutrients like P or K are normally supplied via mineral dissolution in natural systems, nitrogen is in general supplied via fixation of N from the atmosphere (Graham and Vance, 2000). In some ecosystems N supply via rocks might be a relevant source (Houlton et al., 2018; Holloway and Dahlgren, 2002). In general, and specifically under intensified demand scenarios created by enhanced biomass growth rock, rock N supply will not keep up

with the demand.

Many options of carbon dioxide removal rely on the production of biomass (*i.e.*, biochar, afforestation, carbon capture and storage from bio energy (BECCS), bio fuels). These CDR-methods demand, if driven to an optimum, more geogenic nutrients as typically available to plants from the soil-rock-systems in the long-term, specifically in humid, tropical areas, where soils are deeply weathered and show naturally low

nutrient contents (Hamdan and Bumham, 1996) that could not supply an additional intense biomass growth. A study on commercially exploited forests in the U.S. points out that intensive harvest can withdraw more nutrients from the soils than can naturally be resupplied (de Oliveira Garcia et al., 2018).

The intensive withdrawal of nutrients should be included in a framework for biogeochemical cycle management under NET deployment. The withdrawal of each K and P from cropland amounts globally to more than 8 Mt a$^{-1}$ (Suppl. Mat. Fig. S2-2). For many ecosystems the natural resupply and potentially limiting effects under absence of deliberate fertilisation practices, is unknown or merely based on meta-analyses or model studies.

Due to desired global carbon sequestration goals (as in models for afforestation), growth rates will likely be driven to the maximum, which implies an increased demand of nutrients. Models show that N and P limit the global carbon sequestration potential for forests (Goll et al., 2012; Kracher, 2017). Nutrient release by EW can therefore play a relevant role in supporting the high demand. Particular rock classes contain, on average, higher K, P (Fig. 3), or micronutrients like Zn or Se, than others. To ensure a balanced supply of the needed elements, it is therefore necessary to consider not one specific rock type during EW application.

Considering a subtropical weathering scenario in combination with Miscanthus growth for BECCS, acid igneous rocks show a high potential to (partly) resupply extracted potassium, while (ultra-)basic rocks can (partly) resupply phosphorus (Fig. 3). Many earlier studies on EW focussed on dunite to maximize inorganic $CO_2$ sequestration, with the side effect of adding high levels of Ni and Cr to the system (e.g. Schuiling and Krijgsman, 2006; Hangx and Spiers, 2009). Later, basalt was added to the discussion (Beerling et al., 2018; Strefler et al., 2018; Hartmann et al., 2013). It is characterised by an elevated geogenic nutrient supply compared to ultrabasic rocks like dunite (Fig. 3), but still features a sufficiently high inorganic $CO_2$ sequestration potential (Fig. 2, and Strefler et al., 2018). Future application scenarios will likely use a mixture of locally available material to optimize both organic and inorganic carbon storage. Optimising the nutrient composition may come at the price of reducing the inorganic carbon sequestration potential, as some rock types with high nutrient content have low sequestration potentials (Fig. 2). If additional soil properties, like cation exchange capacity, water content/hydrology, and pH, are optimised, this reduction of inorganic carbon sequestration may be compensated by elevated biomass uptake and organic carbon storage.

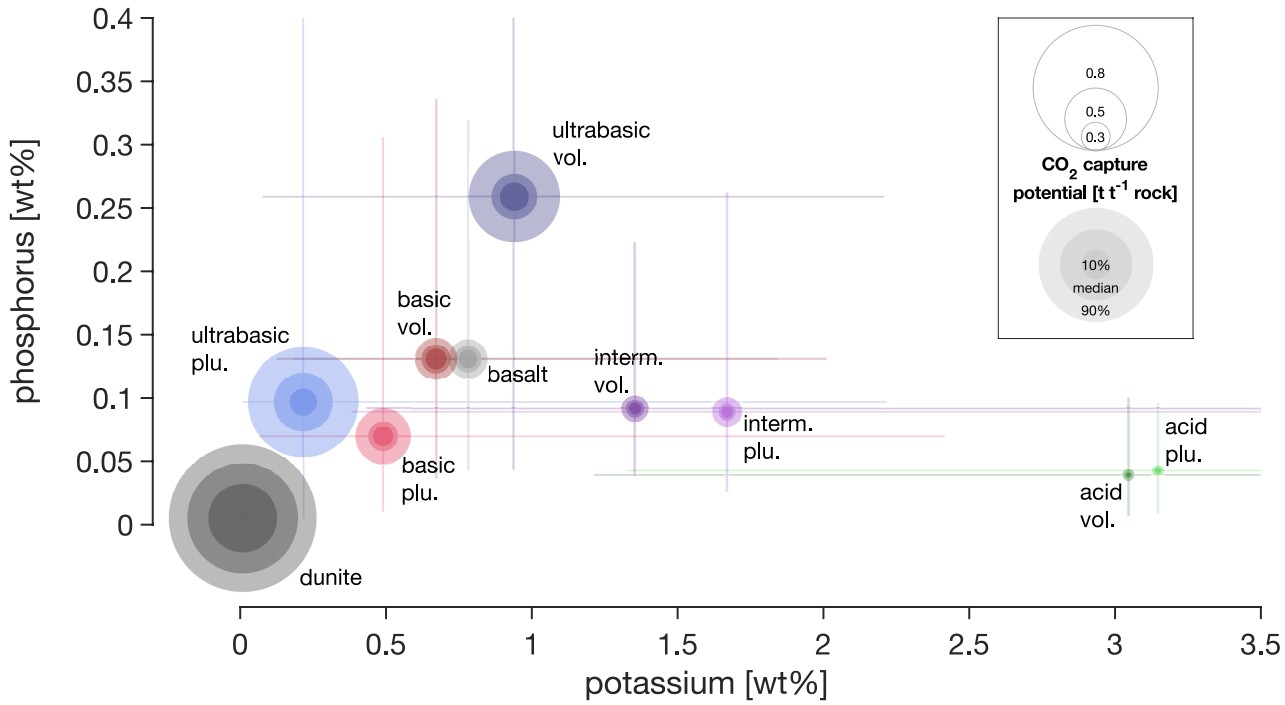

**Fig. 2: The averaged relative K and P contents of igneous rocks (middle point: median, whiskers: 10/90%iles, some cut of at extreme ends for better graphic representation), classified by SiO₂ content (ultrabasic: <45%, basic: 45-52%, intermediate: 52-63%, acid: >63%). The circles indicate their potential to capture CO₂. Statistical data are from the GEOROC database (Sarbas, 2008), details in Suppl. Mat. S3. Documentation on CO₂ capture potential calculation in Suppl. Mat. S4. A map with the global distribution of all classes is available in Supp. Mat. S5. Basalt and dunite were added separately as reference for commonly discussed rock types.**

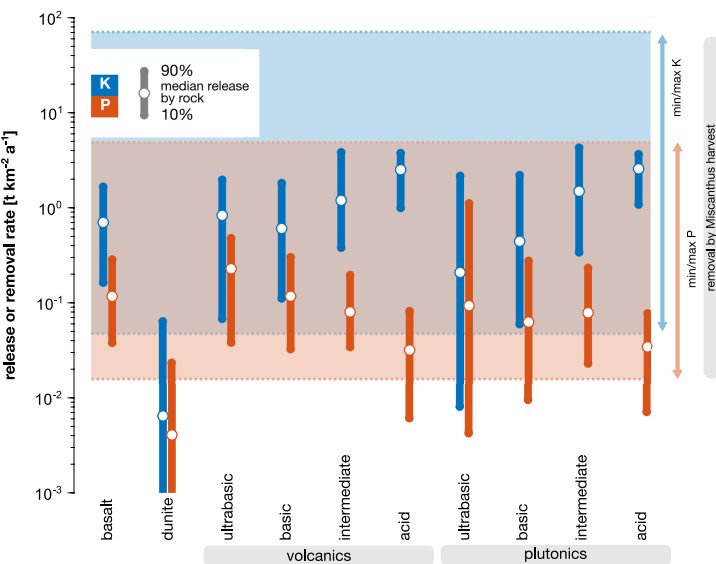

**Fig. 3 Weathering release rates (circles; bars as variability indicator) of P and K from selected rocks (assuming their full dissolution under a natural subtropical weathering scenario) and extraction of those nutrients by harvesting Miscanthus energy grass (blue/red areas indicate range between min. and max. nutrient content of different Miscanthus species multiplied with min. and max. yield reported in Brosse et al. (2012), Suppl. Mat. Tabs. S2-1 and S2-2). Details on rock dissolution and nutrient release rates in Suppl. Mat. S6 and on plant nutrient removal and additional to Miscanthus data on major crops in Suppl. Mat. S2. Dunite values for the 10%ile were cut off for better overall visibility**

The introduction of additional nutrients to the soil system will not necessarily lead to an additional $CO_2$ uptake and increased $CO_2$ sequestration potentials of biomass-based NETs, if enough nutrients are supplied by traditional fertilisation. However, forest areas may benefit from slow release long term available nutrients as they may be less easy to be re-supplied on a regular basis by agrotechnical machinery. Also, industrial fertiliser may be unaffordable in low income regions, thus rock products could replace parts of the fertiliser (Ciceri and Allanore, 2019). A wider adoption of rock product utilisation may also lead to the development of new and optimised application techniques.

**Nutrient retention**

Nutrients released from industrial fertilizers or from natural rock products can be taken up by the plant, washed away, or retained by the properties of the soil. The latter is called retention capacity and is important to store nutrients in a plant available form. It has been shown that the weathering of basaltic material increases the cation exchange capacity, leading to an increased retention of nutrients (Anda et al., 2013, 2015). This is especially important for areas in which nutrients from industrial fertiliser material are quickly washed out, e.g., from the deeply weathered soils (*e.g.*, oxisols) in tropical regions (Leonardos et al., 1987; Ciceri et al., 2017). In such settings, it will be favourable to establish improved soil conditions with optimised nutrient retention.

Another application case is the fertilisation of forests, specifically on areas which are re-forested after agricultural use. With increasing atmospheric $CO_2$ concentrations, an increase in biomass productivity on non-agricultural areas is expected through the $CO_2$ fertilisation effect (e.g. Ciais et al., 1995; Körner et al., 2007; Norby and Zak, 2011), especially with regard to afforestation efforts and general tree growth. This effect has yet to be clearly shown (Leuzinger et al., 2011), and is likely limited by soil fertility (Oren et al., 2001; Bader et al., 2013). It can already be observed that nutrient supply by rock weathering, specifically P, K, Mg, and Ca, can be the limiting factor of tree growth under elevated atmospheric $CO_2$ (Jonard et al., 2015). Woodland soils might be amended with selected minerals or rocks to supply sufficient nutrients to keep up growth under elevated atmospheric $pCO_2$ conditions and organically bind carbon, a scenario that should be explored further for its potential to enlarge affected carbon pools. At some point, depending on the environmental setting, biomass growth will be limited by nutrient supply and as such, model outputs for $CO_2$ sequestration potentials of afforestation are likely to be overestimated, if geogenic nutrient cycles are not included in the assessment, as Goll et al. (2012) have shown exemplary for the C, N, and P cycles using a model.

The $CO_2$ sequestration effect of afforestation is even larger if soil organic carbon changes are taken into consideration: Depending on the underlying lithology, the organic carbon pools can be increased (Li et al., 2017), a process that may be optimised by the spreading of selected rock products.

Overall, specific element deficits in soils need to be mapped, since it can also affect the plant content of valuable, if not essential elements for human nutrition (Zhang et al., 2017; Hengl et al., 2017; White and Zasoski, 1999). It is necessary to be able to predict, which application amounts of elements causes a certain response in the biomass pool above and below ground. Such data are still scarce and inconclusive

(Manning, 2010) but are needed if EW should be used as a method to help manage carbon and nutrient pools.

Biochar is another NET that has a beneficial effect on the retention of nutrients (Fig. 1). Due to its large surface area and increased cation exchange capacity, nutrients can be sustainably retained in soils
(Lehmann, 2007; Liang et al., 2006), effectively saving applied fertiliser (Laird et al., 2010). Increased nutrient retention may increase the overall $CO_2$ sequestration potential of biomass-based NETs through the long-term availability of nutrients. However, the order of magnitude of the effect remains to be shown.

**Soil hydrology**

The availability of water is essential for high crop yields (Rockstrom et al., 2007) and soil hydraulic properties fundamentally steer the availability of water to plants (Bodner et al., 2015; Pinheiro et al., 2019). The soil hydraulic conductivity is a measure of how easy water can percolate though the soil column. It depends largely on the grain size distribution of the soil. Roughly, coarse (sandy) soils have a higher hydraulic conductivity than fine (clayey) soils (Rawls et al., 1982). Spreading large amounts of
rock products with very small grain sizes (silty to clayey) on land potentially leads to a decrease in soil hydraulic conductivity, which may lead to decreased weathering speeds due to local pore water oversaturation or enhanced surface runoff. However, there are some indications that the addition of biochar can be used to control hydraulic conductivity (Masiello et al., 2015; Barnes et al., 2014), which could enable the use of smaller grain sizes for EW, enhancing its potential, which strongly depends on the grainsize
(Strefler et al., 2018).

As another hydraulic property, the water holding capacity determines how much water is kept in the soils and potentially is available to plants. This parameter becomes increasingly important with more frequently appearing droughts due to climate change (Kang et al., 2009). Biochar could be used to improve the water holding capacities of soils (Omondi et al., 2016; Liu et al., 2017), and also increase the plant
available water in some cases (Masiello et al., 2015). This may render dryer regions or areas with unfavourable soil physical properties (Basso et al., 2013) usable for bio energy plants and/or afforestation. There are also indications that improvement of soil hydrology by biochar may increase yield potentials (Akhtar et al., 2014; Xu et al., 2015; Al-Wabel et al., 2018).

It is important to point out that all potential changes of soil physical properties due to biochar application
strongly depend on its type, more specifically the feedstock and production temperature (Gul et al., 2015). The combination of rock product and biochar application however was not addressed in previous research, at all, but may provide a relevant potential to increase and maintain soil carbon.

**Soil pH**

Soil pH steers the availability of elements to plants (Kabata-Pendias, 2010; Loomis and Morris, 1983).
At pH values well below 7, nutrients become less available to plants and potentially harmful trace metals are successively mobilised. Nitrogen-fixing bacteria are also depending on a specific pH (Graham and Vance, 2000). Soil acidification on heavily used cropland is a problem (Helyar and Porter, 1989), which

may lead to a decrease in crop yields. The main reason is the higher mobility of most exchangeable metals at low pH, which decreases logarithmically with increasing pH (Kabata-Pendias, 2010; Robinson et al., 1996; Tack et al., 1996; Harter, 1983). Levels of pH 6 and higher generally ensures very low levels of exchangeable harmful metals, with the exception of arsenic, depending on the oxidation state (Dixit and Hering, 2003). The release of base cations from rock flour leads to a soil pH increase. Studies have demonstrated the effectiveness of basalt powder application in raising the soil pH up to 8 and higher (e.g. Gillman et al., 2001; Nunes et al., 2014). The effect is similar to agricultural liming, which is a common practice to counteract soil acidification on cropland (West and McBride, 2005). Despite the fast dissolution rate of carbonate minerals, they are in general, until today, not considered for EW scenarios, because of possible carbonate precipitation and subsequent $CO_2$ release in the ocean (Hartmann et al., 2013) or due to the potential release of $CO_2$ by excess fertiliser application (Semhi et al., 2000; Perrin et al., 2008). The potential of carbonates in EW strategies remains to be studied, while silicate application is in the focus of recent research (Taylor et al., 2015). It could be a potential economic benefit to replace agricultural lime by silicate rock flour, bearing in mind that silicate dissolution rates are in general several orders of magnitude lower, with strong variability between different minerals (Lasaga, 1995; Brantley et al., 2008). Thus, the efficacy is decreased due to the slower release rate of cations, but other properties like nutrient retention or soil hydrology might be improved (cf. previous chapter). It remains to be investigated how (fast) the termination of pH stabilising silicate rock powder application will affect the soils. If relatively immobile potentially harmful metals accumulate at elevated pH values over the application period, an excessive and harmful release of toxic substances might occur in case of a future drop of pH due to changes in pH controlling minerals, land use or general environmental conditions. Once the deployment of material rich in trace elements of concern is started, it is obligatory to maintain a stabilised pH environment, strengthening the need for material with low harmful trace element concentrations (requirements may differ depending on ecosystem type).

Assuming that pH stabilisation and beneficial changes in soil hydrology (cf. previous section) are achievable by biochar and EW a significant additional $CO_2$ uptake can be expected, based on the effect that soils are made usable for biomass-based NETs, that couldn't support sustainable biomass growth before.

### Soil biota

Chemical weathering of rocks can be significantly mediated by macro- and microbiota (Schwartzman and Volk, 1989; Uroz et al., 2009; Hoffland et al., 2004; Blouin et al., 2013), although the order of magnitude is a matter of debate (Drever, 1994). This is specifically the case for mycorrhizal fungi and microbes, which create physico-chemical conditions that accelerate the dissolution of minerals (Taylor et al., 2015). The weatherability of minerals depends on the type of bioinoculant (Nishanth and Biswas, 2008; Benzerara et al., 2005; Cuadros, 2018), implicating that a supervision and management of the soil microbiota is necessary to optimise both, crop yields and rock weathering. Microbial populations in soils respond to the addition of biochar (Warnock et al., 2007) by providing a refuge for bacteria and fungi (Pietikainen et al., 2000; Saito, 1990), increasing nutrient availability, creating favourable pH conditions and other processes discussed in Lehmann et al. (2011).

Earthworms have been observed to thrive in biochar amended soils (Topoliantz and Ponge, 2005). Increased abundance of earthworms will likely increase bioturbation effects (Carcaillet, 2001; Major et al., 2010), leading to a better distribution of biochar and rock flour in deeper layers of the amended soils, increasing reactive surfaces of mineral grains. Bioturbation might also be a key process to achieve high
$CO_2$ sequestration rates by weathering, as, *e.g.,* Earthworms can enhance mineral weathering (Carpenter et al., 2007; Carpenter et al., 2008) and contribute to the downward transport of added rock products into deeper soil layers (Taylor et al., 2015).

**Trace metals**

Soils are an important sink in the environmental cycling of trace metals (Kabata-Pendias, 1993). Besides
naturally occurring concentrations, depending on the underlying lithology, the major source of trace metals to soils is agricultural practice, leading to an enrichment due to the application of manure, sewage sludge, fertilisers, and pesticides, which all contain metals to a certain extent (Gonnelli and Renella, 2013). Field studies using sewage sludge as fertiliser have shown a marked uptake by the crops and increased mobilisation of trace metals in the runoff water (Alloway, 2013). Adding to the anthropogenic
input, the introduction of additional rock products with elevated levels of trace metals (Fig. 4) could lead to a critical exceedance of environmental thresholds if improper rock material is used due to inconsiderate management. This is, however, relating to the solubility of minerals within the used rock type and the redox and pH conditions. An EW soil incubation experiment using an olivine-rich rock product, with elevated Ni and Cr concentrations in the source material (Fig. 4) showed only a few occurrences of ele-
vated Cr levels but no Ni increase in the aquatic solution compared to a blank treatment, leading to the conclusion that the soil solid phase will be successively enriched in those elements (Renforth et al., 2015). The availability of heavy metals to biota remains an issue of ongoing discussion (Nagajyoti et al., 2010). The main elements of concern in source rocks with the highest sequestration potential (ultramafic rocks) are Ni and Cr. Especially the early discussed dunite application for EW must trigger the discussion about
its high Cr and Ni contents (Fig. 4) and is therefore ruled out for large scale application on cropland.

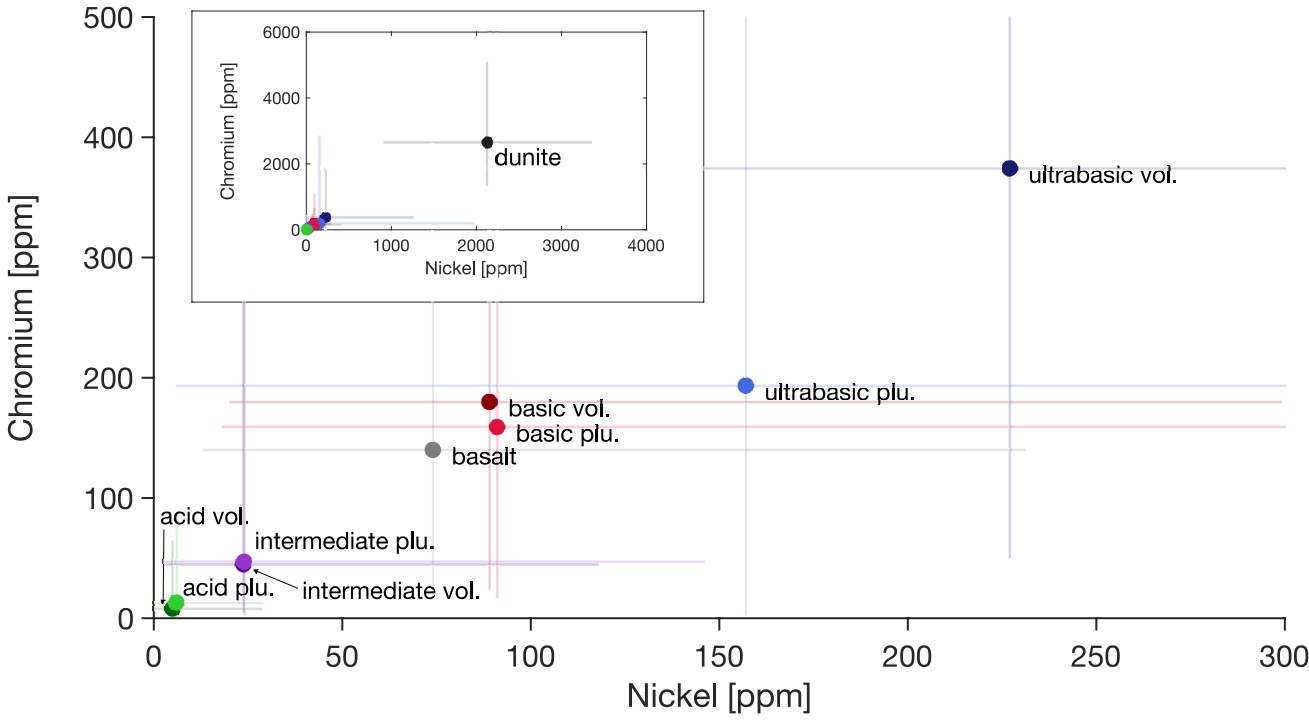

**Fig. 4: Contents of Ni and Cr in igneous rocks, classified by SiO₂ content (cf. Fig. 2). Circles indicate median values, whiskers are the 10th and 90th percentiles. Extreme values of percentiles were cut off for better visibility of data. Extra figure for visualisation of dunite as it features extremely high values. For detailed statistics, cf. Suppl. Mat. S7.**

If an application with rocks of high trace metal concentrations of concern is considered, it is necessary to stabilise the soil pH even after cessation of such actions in order to maintain the fixation of toxic elements, because of the strong pH control on metal mobility. A study of long-term sewage sludge application has shown that the pH had to be stabilised by liming in order to prevent phyto-toxicity of Cu and Zn (McBride

et al., 2004). Additionally, the metal availability to plants has been shown to be influenced by the soil texture, with marked differences for different elements (Qian et al., 1996). This underlines the necessity to control or specifically design the grain size distribution of the soil to control water content, pH and oxygen content. To further ameliorate the issue, biochar, which has been shown to immobilise heavy metals in soils, depending on feedstock and production conditions (Ahmad et al., 2014; Beesley et al.,

2011) could be jointly applied with rock powder. This would mean that potential limitations of fertiliser or rock spreading due to thresholds put in place for environmental protection could be overcome by a sensible management of biochar utilisation. Applying biochar products does not remove elements of concern, but the problem of heavy metal accumulation could be dampened bioremediation through heavy metal accumulating plants (Rajkumar et al., 2012). This in turn could be a potential new source of raw

material for industrial use (Schuiling, 2013), though it is likely not applicable on a global scale, since this would compete against food and energy plant production, which is already an issue (Tilman et al., 2009). The alleviation of trace metal effects does not directly affect CO₂ sequestration rates but could overall increase potential deployment areas for EW.

## Conclusion

Looking forward it is likely that a portfolio of options will be established to optimize the sequestration effect and minimize negative impacts. The combination of previously separately studied NETs to increase the sequestered carbon pool should consider the management of biogeochemical cycles and optimize the combined application of Enhanced Weathering and biochar in context of biomass-based methods like BECCS and afforestation to maximize carbon capture as well as food production. It is therefore essential to address combined effects of NET co-deployment in future research projects.

As all presented interactions take place in the soil, future research should put a focus on creating an optimized soil product for an optimal long-term sustainable carbon management. We propose that research around biomass-based NET interactions becomes the science of artificial soil products, which are most likely created on depleted and degraded soils especially in the sub(tropics). It may consist of the locally available "base soil" mixed with charcoal products to enhance hydraulic properties and nutrient retention, as well as rock powder, which raises the soil pH, provides nutrients and sequesters $CO_2$ at the same time. This engineered and managed soil could increase carbon pools and crop production, while contributing to tackle the issue of climate change. It remains to be studied where suitable material is available at the regional scale (Suppl. Mat. S5). The parameterisation of element release rates permitting a sustainable management are still subject to large uncertainties and the effects of massive rock product spreading will change the soil structure to an extent that remains to be explored.

The introduction of non-authigenic material into the environment, even if of bio- or geogenic origin, will increase the entropy of the system, making it difficult and expensive (energy and economic-wise) to quickly revert back into the "undisturbed" state, once large-scale deployment started. Thus, the continuous deployment of NETs at the global scale in an order of magnitude that would measurably impact atmospheric $CO_2$ levels must be seriously weighed. However, the high probability of NET adoption in the near future makes it imperative to create efficient cooperation networks across all involved disciplines in order to conceive the necessary knowledge on actual $CO_2$ sequestration potentials and century scale global carbon pool changes.

## Competing interests

The authors declare that they have no conflict of interest.

## Author contribution statements

This article was conceived by the joint work of J.H. and T.A. Both participated in discussions, planning and writing, with the lead of T.A..

## Acknowledgments

This research was executed with the financial support of the German Research Foundation's priority program DFG SPP1689 on "Climate Engineering–Risks, Challenges and Opportunities?" and specifically

the CEMICS2 project. Further support came from the Deutsche Forschungsgemeinschaft (DFG) under Germany's Excellence Strategy – EXC 20 2037 'Climate, Climatic Change, and Society' – Project Number: 390683824, contribution to the Center for Earth System Research and Sustainability (CEN) of Universität Hamburg, and from the previous EXC177 'CLISAP2', Universität.

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
