# Peer review of "Ideas and perspectives: Synergies from co-deployment of negative emission technologies"

_Biogeosciences, 2018_

## Referee Comment (RC1) · Anonymous Referee #1 · 25 Dec 2018

General comments: The work of Amann and Hartmann reviews the use of carbon negative technologies, particularly Enhanced Weathering (EW). The main view point expressed by the authors is the necessity for a "co-deployment" of several technologies, because no single approach will be able to reach the 1.5C target set by the IPCC. The manuscript is timely, and I support the overall goal. However, the manuscript is largely qualitative rather than quantitative, and seems to focus on soil and EW, whereas the title, and partially the abstract, implied a much broader review. In my opinion this work can be published in BG, but some amendments are necessary.

Specific comments: - Figure 1 is not sufficiently clear. I would place the benefactor on

top of the beneficiaries. Also I was not sure how to read the figure. For example, in the first block EW enhances the nutrient pool and maintain afforestation? I was not sure about the meaning of benefactor and beneficiary here;

- P.2, line 11. I would appreciate if the authors would insert also a comment on the social and environmental effect of mining;

- P.3, line 13 introduces the content of enhanced weathering but not all readers are familiar with what this is;

- P.3, line 26 mention for the first time dunite and basalt. Can the author specify why they choose these specific examples? References are reported but the reader is left wondering what's special about these rocks;

- Figures 2-3 are interesting, but I am wondering about the overall availability of these resources. As an example, I am not sure about the relevance of komatiite. I understand this is an explicative diagram, but the context here is that of global-deployable technologies. I feel the text should explain better the abundance and distribution of some key resources, or at least provide relevant references/tabula data;

- P.5., line 20 would benefit of a reference. Current models do not consider nutrient availability?;

- The paragraph on soil hydrology does not specify the size of the grains that would decrease soil conductivity. I understand this would depend on the soil but can at least a range be specified? It would be important to specify size ranges for these rock flours;

- The paragraph on soil biota is extremely qualitative. At least one could point out which minerals are more susceptible to bioweathering, or which structural elements are more needed by soil bacteria. It would also be important to point out the interaction between type of crops and bacteria. In fact I was surprised not to see a paragraph dedicated on how different crops may work synergistically with different type of rocks. The entire manuscript seem to focus on afforestation, but it would be interesting to point out how

agriculture may also benefit from EW;

- A general comment on nitrogen would benefit the manuscript (e.g., Nitrogen in rock: Occurrences and biogeochemical implications 2002 JoAnn and M. Holloway).

Technical comments: - Figure 1 caption can be improved. Land-based should be spelled with a dash and capital letters should be double-checked;

- I may have missed it but I do not think EW was ever defined;

- I would switch the phrasing of the title to: "Synergies from co-deployment of negative emission technologies: Ideas and perspectives";

- Latinism such as e.g. and i.e. should be italicized;

- I would change the heading of the last paragraph from "synthesis" to "summary" or "conclusion";

―――――――――――――――――

---

## Referee Comment (RC2) · Anonymous Referee #2 · 18 Jan 2019

The authors present an overview of how different negative emission technologies might interact and thereby trigger additional carbon uptake. The work is timely and important as such synergies have not been addressed so far in great detail. However, the authors focus on the beneficial effects of enhanced weathering and biochar on afforestation and BECCS while the title suggests a somewhat broader overview. Additionally, some statements would benefit from describing synergies in a more quantitative way. I think the manuscript can be published in Biogeosciences after some changes. The study will encourage further research about the interactions between different types of negative emission technologies.

[Figure]

Specific comments:

- Title: As reviewer #1 I would also move "ideas and perspectives" to the end of the title. Maybe "Synergies from co-deployment of land-based negative emission technologies: Ideas and perspectives" to clarify this paper is about soil/land-based strategies.

- P2, line 1: "of" is repeated three times. Suggestion: "assessing the effects of combined...".

- P2, line8: "e.g." implies that there are more nutrients so you can remove "and others".

- I agree with reviewer #1 that it's not clear at all how to read Figure 1. Intuitively I would expect benefactors on top of beneficiaries and additional CO2 sequestration. I was also confused by the two verbs connecting benefactors and beneficiaries. Lastly, the additional CO2 sequestration is unclear. I assume the downward arrows mean that e.g. EW increases CO2 sequestration via BECCS or AFF (flux from the atmosphere to the land) but one could also interpret it as a decline.

- P3, line 3: envisions.

- P3, line 21: But only if these new forests are harvested.

- P3, line 25: Higher than other rock types?

- P3, line 29: types

- Figure 2: I think it would be interesting to show the CO2 capture potential of dunite in this figure as it seems to be a highly relevant rock. After all you show Komatiite which also has very low K and P contents.

- Figure 3: So the extraction range was derived from min/max nutrient contents but what yields were assumed for Miscanthus (range is 40-4400 t/km2 according to Table S5.-1)? In addition, Figure 3 seems to not be referred to in the text.

- P4, line 3: I think this isolated sentence would fit better in the second paragraph of

this section.

- P5, line 14: nutrients limit tree growth?

- P6, line 17: Confusing, split this sentence into two sentences.

- P6, line 24: Reference missing in the reference list.

- P7, line3: "where considered to be of concern" sounds awkward, I think it can be removed.

- P7, line 10: Reference seems to be at the wrong position.

- Table 1: The reader is left wondering what values are typical. Can you also provide numbers for some other rock types for comparison?

---

## Author Comment (AC1) · 12 Apr 2019

Dear reviewer, Thank you for a constructive and thoughtful review. We acknowledge your concerns and will incorporate all suggestions in the revision. We like to specifically address a few issues raised.

**Reviewers comment**

Our reply

**However, the manuscript is largely qualitative rather than quantitative, and seems to focus on soil and EW, whereas the title, and partially the abstract, implied a much broader review.**

This point is well taken. The synergies mainly apply to soil-based NETs, maybe we can reflect this in the title. However, "soil-based NET" is not a term established as such in the community. About the qualitativeness: This is a good point. There is some data out there on the processes mentioned, however the framework of research is very often so far off the focus of our manuscript, that it would introduce a level of detail that is misguiding for the purpose of this manuscript. We explicitly chose the format of a "perspectives piece" to identify the main important processes that need to be considered in future research on NETs and their combined effects. This should provide a guideline for projects to come. Yet, we try to be at least a little more specific in the discussion of the processes.

**- P.3, line 26 mention for the first time dunite and basalt. Can the author specify why they choose these specific examples? References are reported but the reader is left wondering what's special about these rocks;**

We extended the text to explain the background and we will add some broader categories of rock geochemistry (see comment below) to be more general in our arguments.

**- Figures 2-3 are interesting, but I am wondering about the overall availability of these resources. As an example, I am not sure about the relevance of komatiite. I understand this is an explicative diagram, but the context here is that of global-deployable technologies. I feel the text should explain better the**

**abundance and distribution of some key resources, or at least provide relevant references/tabula data;**

We will remove the very specific selection of rocks, which was chosen as available in the database. We generalized the data now, by distinguishing classes via SiO2 content of volcanic and plutonic ultrabasic/basic/intermediate/acid rocks. This classification enables us to give a broader and more general overview of what to look for in a rock. Additionally, we add the rock types dunite and basalt as commonly discussed types for reference. A map with the global distribution of the distinguished classes will be provided in the supplement. As an example, basic volcanic rocks (this class contains basaltic rocks) covers about 3.5

---

## Author Comment (AC2) · 12 Apr 2019

Dear reviewer, Thank you for a constructive and thoughtful review. We acknowledge your concerns and will incorporate all suggestions in the revision. We like to specifically address a few issues raised.

**Reviewers comment**

Our reply

[Figure]

**However, the authors focus on the beneficial effects of enhanced weathering and biochar on afforestation and BECCS while the title suggests a somewhat broader overview.**

In this respect, you and your co-reviewer raise the same concern. We acknowledge this and will adapt the text, and maybe even the title to be more specific about the NETs discussed. Please look into our reply to reviewer 1 for an extended answer.

**Additionally, some statements would benefit from describing synergies in a more quantitative way.**

Also, here the concern is also raised by reviewer 1. We explicitly chose the format of a "perspectives piece" to identify the main important processes that need to be considered in future research on NETs and their combined effects. This should provide a guideline for projects to come. Yet, we try to be at least a little more specific in the discussion of the processes.

**- I agree with reviewer 1 that it's not clear at all how to read Figure 1. Intuitively I would expect benefactors on top of beneficiaries and additional CO2 seques- tration. I was also confused by the two verbs connecting benefactors and beneficiaries. Lastly, the additional CO2 sequestration is unclear. I assume the downward arrows mean that e.g. EW increases CO2 sequestration via BECCS or AFF (flux from the atmosphere to the land) but one could also interpret it as a decline.**
As you both had similar issues with this figure, we will modify it to have it more logically structured and less room for interpretation.
**- Figure 2: I think it would be interesting to show the CO2 capture potential of dunite in this figure as it seems to be a highly relevant rock. After all you show Komatiite which also has very low K and P contents.**

As mentioned in the caption, dunite contains so little K and P that it wouldn't be easy to visualize. We will try to come up with a solution for this.

**- Table 1: The reader is left wondering what values are typical. Can you also provide numbers for some other rock types for comparison?**

We generalized the data now, by distinguishing classes via SiO2 content of volcanic and plutonic ultrabasic/basic/intermediate/acid rocks. This classification enables us to give a broader and more general overview of what to look for in a rock. Additionally, we add the rock types dunite and basalt as commonly discussed types for reference. Due to the increased amount of data, we may convert the table into a figure similar to Fig. 2.

---

## Author Response (AR1)

**Author's response**

Again, we thank all reviewers for their careful reviews. We acknowledge the issues brought up and, in the following, we reply to all of them point-by-point.

**Reviewers comment**
Our reply

**Anonymous Referee #1**

**The work of Amann and Hartmann reviews the use of carbon negative technologies, particularly Enhanced Weathering (EW). The main view point expressed by the authors is the necessity for a "co-deployment" of several technologies, because no single approach will be able to reach the 1.5C target set by the IPCC. The manuscript is timely, and I support the overall goal.**

**However, the manuscript is largely qualitative rather than quantitative, and seems to focus on soil and EW, whereas the title, and partially the abstract, implied a much broader review. In my opinion this work can be published in BG, but some amendments are necessary.**

This point is well taken. The synergies mainly apply to soil-based NETs, maybe we can reflect this in the title. However, "soil-based NET" is not a term established as such in the community. About the qualitativeness: This is a good point. There is some data out there on the processes mentioned, however the framework of research is very often so far off the focus of our manuscript, that it would introduce a level of detail that is misguiding for the purpose of this manuscript. We explicitly chose the format of a "perspectives piece" to identify the main important processes that need to be considered in future research on NETs and their combined effects. This should provide a guideline for projects to come. Yet, we try to be at least a little more specific in the discussion of the processes.

*Specific comments*

**- Figure 1 is not sufficiently clear. I would place the benefactor on top of the beneficiaries. Also I was not sure how to read the figure. For example, in the first block EW enhances the nutrient pool and maintain afforestation? I was not sure about the meaning of benefactor and beneficiary here;**

We changed the entire figure on different levels. Benefactor/beneficiary were replaced by something more straightforward. The "connecting verbs" were amended with arrow boxes to indicate the reading direction. The $CO_2$ sequestration effect symbology was changed to avoid misunderstanding it as $CO_2$ release.

**- P.2, line 11. I would appreciate if the authors would insert also a comment on the social and environmental effect of mining;**

A short remark was added, to address the potential after-use of mines. The social effect was not address due to the fact that it touches a whole different topic.

**- P.3, line 13 introduces the content of enhanced weathering but not all readers are familiar with what this is;**

A brief explanation was added.

**- P.3, line 26 mention for the first time dunite and basalt. Can the author specify why they choose these specific examples? References are reported but the reader is left wondering what's special about these rocks;**

We extended the text to explain the background and we added some broader categories of rock geochemistry (see comment below) to be more general in our arguments.

**- Figures 2-3 are interesting, but I am wondering about the overall availability of these resources. As an example, I am not sure about the relevance of komatiite. I understand this is an explicative diagram, but the context here is that of global-deployable technologies. I feel the text should explain better the abundance and distribution of some key resources, or at least provide relevant references/tabula data;**

We removed the very specific selection of rocks, which was chosen as available in the database. We generalized the data now, by distinguishing classes via $SiO_2$ content of volcanic and plutonic ultrabasic/basic/intermediate/acid rocks. This classification enables us to give a broader and more general overview of what to look for in a rock. Additionally, we added the rock types dunite and basalt as commonly discussed types for reference. A map with the global distribution of the distinguished classes is now provided in the supplement (S5). As an example, basic volcanic rocks (this class contains basaltic rocks) covers about 3.5% of the land surface, according to the latest global lithologic map (Hartmann & Moosdorf 2012).

**- P.5., line 20 would benefit of a reference. Current models do not consider nutrient availability?;**

A reference was added, showing the importance of P cycling, which is commonly not included.

**- The paragraph on soil hydrology does not specify the size of the grains that would decrease soil conductivity. I understand this would depend on the soil but can at least a range be specified? It would be important to specify size ranges for these rock flours;**

While it isn't possible to be extremely specific, we added a sentence on how clayey, silty, and sandy grainsizes are expected to change hydrology.

**- The paragraph on soil biota is extremely qualitative. At least one could point out which minerals are more susceptible to bioweathering, or which structural elements are more needed by soil bacteria.**

We added a sentence on how weatherability of certain minerals depends on the type of bioinoculant.

**It would also be important to point out the interaction between type of crops and bacteria. In fact I was surprised not to see a paragraph dedicated on how different crops may work synergistically with different type of rocks.**

As this is a manuscript with a somewhat broad perspective, we deliberately did not go into details of synergies between crops and bacteria as the focus on NETs would be lost. However, we included the relationship between bacteria and weathering and crops and weathering.

**The entire manuscript seem to focus on afforestation, but it would be interesting to point out how agriculture may also benefit from EW;**

We do not fully understand this part of the comment. The manuscript approaches the synergies specifically with a focus on products released from weathering, while afforestation is merely addressed as a benefiting NET. How agriculture benefits from EW is specifically addressed in

the sections on nutrient release and retention. If we missed the connection to soil biota here, we kindly ask to be more specific.

**- A general comment on nitrogen would benefit the manuscript (e.g., Nitrogen in rock: Occurrences and biogeochemical implications 2002 JoAnn and M. Holloway).**

Rock N does play a minor role in agricultural systems. A remark on weathering derived N was added.

*Technical comments*

**- Figure 1 caption can be improved. Land-based should be spelled with a dash and capital letters should be double-checked;**

Checked and corrected.

**- I may have missed it but I do not think EW was ever defined;**

We took care of this added defined EW once in the beginning.

**- I would switch the phrasing of the title to: "Synergies from co-deployment of negative emission technologies: Ideas and perspectives";**

We agree and like it better that way, however, the journals rules seem to demand it the other way round: https://www.biogeosciences.net/about/manuscript_types.html: "Manuscripts of this type should be short (a few pages only). The manuscript title must start with "Ideas and perspectives:"." We would be happy to change it, if allowed by the editor/journal.

**- Latinism such as e.g. and i.e. should be italicized;**

Agreed & changed.

**- I would change the heading of the last paragraph from "synthesis" to "summary" or "conclusion";**

Agreed & renamed to conclusion.

**Anonymous Referee #2**

**The authors present an overview of how different negative emission technologies might interact and thereby trigger additional carbon uptake. The work is timely and important as such synergies have not been addressed so far in great detail.**

**However, the authors focus on the beneficial effects of enhanced weathering and biochar on afforestation and BECCS while the title suggests a somewhat broader overview.**

In this respect, you and your co-reviewer raise the same concern. We acknowledge this and will adapt the text, and maybe even the title to be more specific about the NETs discussed. Please look into our reply to reviewer #1 for an extended answer.

**Additionally, some statements would benefit from describing synergies in a more quantitative way.**

Also, here the concern is also raised by reviewer #1. We explicitly chose the format of a "perspectives piece" to identify the main important processes that need to be considered in future research on NETs and their combined effects. This should provide a guideline for projects to come. Yet, we try to be at least a little more specific in the discussion of the processes.´

*Specific comments*

**- Title: As reviewer #1 I would also move "ideas and perspectives" to the end of the title. Maybe "Synergies from co-deployment of land-based negative emission technologies: Ideas and perspectives" to clarify this paper is about soil/land-based strategies.**

Agreed, but difficult as described above in the answer to Reviewer #1.

**- P2, line 1: "of" is repeated three times. Suggestion: "assessing the effects of combined. . .".**

Agreed and changed.

**- P2, line8: "e.g." implies that there are more nutrients so you can remove "and others".**

Agreed and removed.

**- I agree with reviewer #1 that it's not clear at all how to read Figure 1. Intuitively I would expect benefactors on top of beneficiaries and additional CO2 sequestration. I was also confused by the two verbs connecting benefactors and beneficiaries. Lastly, the additional CO2 sequestration is unclear. I assume the downward arrows mean that e.g. EW increases CO2 sequestration via BECCS or AFF (flux from the atmosphere to the land) but one could also interpret it as a decline.**

As you both had similar issues with this figure, we modified it to have it more logically structured and less room for interpretation (also see answer to reviewer 1).

**- P3, line 3: envisions.**

✓

**- P3, line 21: But only if these new forests are harvested.**

Added a clause to be more specific.

**- P3, line 25: Higher than other rock types?**

Yes, this part was amended to complete the comparison.

**- P3, line 29: types**

✓

**- Figure 2: I think it would be interesting to show the CO2 capture potential of dunite in this figure as it seems to be a highly relevant rock. After all you show Komatiite which also has very low K and P contents.**

As mentioned in the caption, dunite contains so little K and P that it wouldn't be easy to visualize. We added it now along with basalt and all the rock classes differentiated by SiO2 content. For reasons of visibility, the axes had to be spread apart.

**- Figure 3: So the extraction range was derived from min/max nutrient contents but what yields were assumed for Miscanthus (range is 40-4400 t/km2 according to Table S5.-1)?**

The minima of nutrient content and yield were multiplied to get the total minimum nutrient removal. Accordingly, the maximum was calculated. An explanation was added to the caption. Also a description was added next to the areas, to directly indicate their meaning.

**In addition, Figure 3 seems to not be referred to in the text.**

Good point. It is now.

**- P4, line 3: I think this isolated sentence would fit better in the second paragraph of this section.**

Yes. We moved it there.

**- P5, line 14: nutrients limit tree growth?**

Acknowledged. We reformulated the sentences to be precise. Geogenic nutrient supply can be the limiting component in an elevated $CO_2$ system.

**- P6, line 17: Confusing, split this sentence into two sentences.**

Agreed and rewritten.

**- P6, line 24: Reference missing in the reference list.**

Fixed.

**- P7, line3: "where considered to be of concern" sounds awkward, I think it can be removed.**

Agreed & removed.

**- P7, line 10: Reference seems to be at the wrong position.**

Agreed & corrected.

**- Table 1: The reader is left wondering what values are typical. Can you also provide numbers for some other rock types for comparison?**

We generalized the data now, by distinguishing classes via $SiO_2$ content of volcanic and plutonic ultrabasic/basic/intermediate/acid rocks. This classification enables us to give a broader and more general overview of what to look for in a rock. Additionally, we add the rock types dunite and basalt as commonly discussed types for reference. Due to the increased amount of data, we converted the table into a xy-plot, using median and 10/90 percentile data. The numerical data is provided in the supplement (S7).

*In the following, we provide a detailed overview over all changes that have been done compared to the original document. It is document comparison generated by Microsoft Word.*

[revised manuscript text omitted]

**Seite 1: [1] Formatvorlagendefinition**                    revised                    01.05.19 20:33:00

EndNote Bibliography: Englisch (Vereinigtes Königreich), Zeilenabstand: einfach

| Seite 1: [1] Formatvorlagendefinition | revised | 01.05.19 20:33:00 |
|---|---|---|

EndNote Bibliography: Englisch (Vereinigtes Königreich), Zeilenabstand: einfach

| Seite 1: [2] hat formatiert | revised | 01.05.19 20:33:00 |
|---|---|---|

Englisch (Vereinigtes Königreich)

| Seite 1: [2] hat formatiert | revised | 01.05.19 20:33:00 |
|---|---|---|

Englisch (Vereinigtes Königreich)

| Seite 1: [3] hat formatiert | revised | 01.05.19 20:33:00 |
|---|---|---|

Englisch (Vereinigtes Königreich)

| Seite 1: [3] hat formatiert | revised | 01.05.19 20:33:00 |
|---|---|---|

Englisch (Vereinigtes Königreich)

| Seite 1: [4] hat formatiert | revised | 01.05.19 20:33:00 |
|---|---|---|

Englisch (Vereinigtes Königreich)

| Seite 1: [4] hat formatiert | revised | 01.05.19 20:33:00 |
|---|---|---|

Englisch (Vereinigtes Königreich)

| Seite 10: [5] hat gelöscht | revised | 01.05.19 20:33:00 |
|---|---|---|

---

## Author Response (AR2)

**Author's response**

We thank the associate editor, Anja Ramming, for her final remark on Figure 1. We changed the appearence, so that the "$CO_2$ effect" is placed at the bottom of the figure now.